# In Situ Photodeposition of Cobalt Phosphate (CoH_x_PO_y_) on CdIn_2_S_4_ Photocatalyst for Accelerated Hole Extraction and Improved Hydrogen Evolution

**DOI:** 10.3390/nano13030420

**Published:** 2023-01-19

**Authors:** Jiachen Xu, Qinran Li, Dejian Sui, Wei Jiang, Fengqi Liu, Xiuquan Gu, Yulong Zhao, Pengzhan Ying, Liang Mao, Xiaoyan Cai, Junying Zhang

**Affiliations:** 1School of Materials Science and Physics, China University of Mining and Technology, Xuzhou 221116, China; 2School of Safety Engineering, China University of Mining and Technology, Xuzhou 221116, China; 3School of Physics, Beihang University, Beijing 100191, China

**Keywords:** photocatalysis, hydrogen evolution, hole transfer, charge separation

## Abstract

The ternary metal sulfide CdIn_2_S_4_ (CIS) has great application potential in solar-to-hydrogen conversion due to its suitable band gap, good stability and low cost. However, the photocatalytic hydrogen (H_2_) evolution performance of CIS is severely limited by the rapid electron–hole recombination originating from the slow photogenerated hole transfer kinetics. Herein, by simply depositing cobalt phosphate (CoH_x_PO_y_, noted as Co-Pi), a non-precious co-catalyst, an efficient pathway for accelerating the hole transfer process and subsequently promoting the H_2_ evolution reaction (HER) activity of CIS nanosheets is developed. X-ray photoelectron spectroscopy (XPS) reveals that the Co atoms of Co-Pi preferentially combine with the unsaturated S atoms of CIS to form Co-S bonds, which act as channels for fast hole extraction from CIS to Co-Pi. Electron paramagnetic resonance (EPR) and time-resolved photoluminescence (TRPL) showed that the introduction of Co-Pi on ultrathin CIS surface not only increases the probability of photogenerated holes arriving the catalyst surface, but also prolongs the charge carrier’s lifetime by reducing the recombination of electrons and holes. Therefore, Co-Pi/CIS exhibits a satisfactory photocatalytic H_2_ evolution rate of 7.28 mmol g^−1^ h^−1^ under visible light, which is superior to the pristine CIS (2.62 mmol g^−1^ h^−1^) and Pt modified CIS (3.73 mmol g^−1^ h^−1^).

## 1. Introduction

The overuse of fossil fuels has led to increased environmental pollution and severe energy shortages [1,2,3]. The development and utilization of green and renewable energy have always attracted extensive attention from researchers [4,5]. Hydrogen (H_2_) is considered one of the most promising clean energy sources because of its high combustion calorific value [6,7]. Since it was first discovered by Fujishima and Honda in 1972 [8], photocatalytic H_2_ evolution through water splitting as an important method has caused extensive exploration due to its eco-friendliness and low cost [9,10,11]. So far, various semiconductor photocatalysts such as oxides [12,13,14], sulfides [15,16,17,18] and oxynitrides [19,20] have been employed for photocatalytic splitting of water. Unfortunately, most of the reported semiconductors bear either limited spectral absorption range or low photostability, which do not meet all the requirements of effective photocatalyst [21,22,23]. The ternary metal sulfide CdIn_2_S_4_ (CIS), as a member of AB_2_X_4_ family, is considered a promising photocatalyst due to its superior physicochemical stability, narrow bandgap, and suitable band position for hydrogen evolution reaction (HER) [24,25,26,27]. However, the practical application of CIS is extremely confined by lack of catalytic active sites and difficulty of hole extraction [28,29,30,31].

Previous studies on semiconductor photocatalysts have shown that co-catalyst modification is a productive strategy to assist photocatalytic performance. The utilization of suitable co-catalysts helps to inhibit electron–hole recombination, accelerate the migration of photogenerated carriers, and provide active sites for HER [32]. Precious metals (Pt, Au, etc.) are the most commonly used co-catalysts for photocatalytic H_2_ production [12,33,34,35]. Considering the high price of noble metal, the introduction of non-noble metal co-catalysts on CIS surface has gradually become a research hotspot. For example, Ma et al. reported that MoP co-catalysts could serve as reactive sites and reduce the energy barrier of CIS for elevated photocatalytic H_2_ evolution [36]. Li et al. synthesized a noble metal-free Co_9_S_8_/CIS photocatalyst, in which Co_9_S_8_ mainly captured and transported photoexcited electrons of CIS to enhance photocatalytic activity [37]. Nevertheless, the disadvantages of synthetic difficulty and frangibility to oxidation limit the large-scale implementation of metal phosphide and sulfide co-catalysts in photocatalytic H_2_ production.

It has been reported that the transport process of photogenerated holes from bulk to surface is the rate-determining step of the photocatalytic water splitting reaction of sulfides and nitrides [38,39]. Cobalt phosphate (CoH_x_PO_y_, noted as Co-Pi) has been extensively studied as an efficient oxygen evolution reaction (OER) accelerator for several types of semiconductor photocatalysts [40,41]. For instance, Lee et al. reported the enhanced photocatalytic H_2_ production of g-C_3_N_4_ in the presence of Co-Pi [42]. Jiang et al. reported that the HER activity of ZnIn_2_S_4_ was significantly promoted by means of photodeposition of Co-Pi [43]. In the presence of sacrificial agent, the photogenerated holes of ZnIn_2_S_4_ were extracted by Co-Pi, followed by injecting into the sacrificial agent, where Co-Pi served as oxidative reaction sites to eliminate the surface hole trap of ZnIn_2_S_4_, reducing the recombination of electrons and holes prior to HER process [43]. Motivated by the aforementioned discussion, we expect that Co-Pi can promote charge separation as well as H_2_ release dynamics of CIS for photocatalysis improvement.

In this work, we reported the enhancement of photocatalytic H_2_ evolution performance of ultrathin CIS nanosheets by using the OER co-catalyst Co-Pi. A simple photochemical approach was employed to deposit amorphous Co-Pi on the surface of CIS nanosheets. The Co atoms in Co-Pi bonded with the unsaturated S atoms of CIS to form Co-S bonds, which improved the hole extraction ability and thus accelerated the hole transfer kinetics from CIS bulk phase to the surface, thereby improving the HER activity. As a result, the photocatalytic H_2_ production rate of Co-Pi/CIS was 7.28 mmol g^−1^ h^−1^, which was 1.95 times that of Pt/CIS. Finally, combined with a series of spectroscopy and photoelectrochemical characterization, the photocatalytic mechanism of Co-Pi/CIS was proposed.

## 2. Materials and Methods

### 2.1. Chemicals

CdSO_4_·8/3H_2_O (AR) was purchased from Sinopharm Group Chemical Reagent Co., Ltd., Shangai, China. Thioacetamide (TAA), InCl_3_·4H_2_O (99.0%), Na_2_HPO_4_ (99.99%), and NaH_2_PO_4_ (99.0%) were purchased from Rhawn Chemical Reagent Company, Ltd., Shangai, China. Co(NO_3_)_2_·6H_2_O (99.99%) were bought from Shanghai Macklin Biochemical Technology Co., Ltd., Shangai, China. All the above chemical reagents were employed as received without further treatment. Nafion (Sigma-Aldrich, St. Louis, MO, USA) was used as a paste for the working electrode preparation. Deionized (DI) water used throughout all the experiments was purified through a Millipore system (18.2 MΩ·cm).

### 2.2. Synthesis of CIS

Amounts of 1 mmol CdSO_4_·8/3H_2_O, and 2 mmol InCl_3_·4H_2_O were added into a solution containing 10 mL ethylene glycol and 50 mL deionized water. After vigorous stirring for 30 min, 4 mmol TAA was added to the mixture followed by vigorous stirring for another 30 min. The mixture was transferred to a 100 mL Teflon-lined autoclave and vigorously stirred, and then the hydrothermal reaction was conducted at 120 °C for 12 h in an electric oven. The final product was collected by centrifuging and washing with ethanol and deionized water three times until the organic residuals were completely removed, and then drying in vacuum for 3 h.

### 2.3. Synthesis of Co-Pi/CIS Composites

Co-Pi was deposited on the surface of CIS nanosheets according to the reported photodeposition method [43]. Typically, 100 mg CIS nanosheets were added to 15 mL Co-Pi (pH = 7) buffer solution containing 0.5 mM cobalt nitrate 0.1 M phosphates with a duration of 2 h under a 420 nm cutoff filter of 500 W Xe lamp.

### 2.4. Materials Characterization

The phase structure of samples was identified using Bruker D8 Advance X-ray diffract meter (XRD, BRUKER Co., Karlsruhe, Germany) with Cu Kα1 radiation (k = 0.15418 nm). The morphology of the as-prepared samples was studied with a Tecnai-G2-F20 transmission electron microscopy (TEM, FEI Co., Hillsboro, OR, USA) and Hitachi New Generation SU8220 field emission scanning electron microscopy (FESEM, HITACHI Co., Kyoto, Japan). The ultraviolet-visible diffuse reflection spectroscopy (UV-vis DRS) was measured by the Cary 300 UV-vis spectrophotometer (Varian Co., Palo Alto, CA, USA). X-ray photoelectron spectroscopy (XPS) data were collected from the K-Alpha photoelectron spectroscope (Thermo Fisher Scientific, Waltham, MA, USA) with monochromatic Al Ka radiation (200 W). Edinburgh FS5 fluorescence lifetime spectrophotometer was used to obtain the stable state photoluminescence (PL) spectra and time-resolved photoluminescence (TRPL) spectra (Edinburgh Instruments Co., Edinburgh, UK). The electron paramagnetic resonance (EPR) spectra were obtained using an EPR spectro-meter (E300, BRUKER Co., Karlsruhe, Germany) with a modulation amplitude of 1 G and microwave power of 3.99 mW.

### 2.5. Photocatalytic Performance Evaluation

The photocatalytic performance was tested under visible light by using a 500 W Xenon lamp as a light source filtered with a nominal 420 nm cutoff filter. In a typical photocatalytic reaction, 10 mg of the photocatalyst was dispersed in 15 mL of methanol aqueous (20 vol%) solution. The mixture was stirred constantly and cooled by external circulating cooling water during the photocatalysis process. The evolved H_2_ amount was analyzed using a gas chromatographer (Shimadzu GC-2030) equipped with a thermal conductivity detector and the high purity nitrogen as carrier gas. 

The apparent quantum efficiency (AQE) was calculated by the following formula: AQE = 2 × number of evolved H_2_ molecules/total number of photons × 100%.

### 2.6. Photoelectrochemical (PEC) Measurements

The photocurrent vs. time (I-t) curves, and Mott-Schottky (M-S) plots were elucidated by the electrochemical workstation (CHI 660E, Chenhua Co., Shanghai, China). A typical three-electrode system includes a working electrode, a reference electrode of saturated calomel electrode (SCE), and a counter electrode of Pt wire. Xenon lamp was utilized as the irradiation source in the PEC measurements with an intensity of 100 mW cm^−2^. The transient photocurrent response was evaluated under visible-light irradiation (the interval is 10 s for light on and off). The preparation process of the working electrode was as follows: 10 mg catalyst was dispersed in 20 µL ethanol, 20 µL deionized water and 10 µL 5% Nafion solution by sonicating for 15 min. Afterwards, 5 µL suspension was deposited on the Fluorine-doped Tin Oxide (FTO) substrate (1 × 2 cm^2^) and dried at room temperature.

## 3. Results and Discussion

XRD was used to study the phase and crystal structure of CIS and Co-Pi/CIS photocatalysts. As shown in Figure 1a, all diffraction peaks can be assigned to cubic phase CdIn_2_S_4_ (JCPDS No. 27-0060) [44]. Diffraction peaks at 27.25° and 47.41° correspond to the (311) and (400) planes of CIS, respectively [45]. Due to the amorphous nature and small amount of Co-Pi, the XRD pattern of Co-Pi/CIS composite is basically consistent with that of CIS [43]. The nanostructure of the samples was characterized by SEM and TEM. From Figure 1b, c, it can be seen that CIS possesses a two-dimensional sheet-like structure with a size of 30–50 nm, and its topography is basically unchanged after loading Co-Pi. Figure 1d is the TEM image of Co-Pi/CIS, showing the small lateral dimension of ~50 nm and ultrathin thickness of ~5 nm of CIS. However, the presence of Co-Pi cannot be identified under these circumstances. Further observation by high-resolution TEM (HRTEM) shows that the lattice fringes with plane spacing of 0.313 nm and 0.327 nm in Figure 1e closely match the (222) and (311) planes of the cubic CIS, respectively. Some fuzzy and disordered dark spots can be observed on CIS surface, which may be assigned to amorphous Co-Pi clusters. Moreover, Pt nanoparticles were loaded on CIS (Pt/CIS) as a comparison, through a similar photodeposition process. Figure 1f is the TEM image of Pt/CIS, indicating that Pt nanoparticles are evenly deposited on the surface of CIS nanosheets. In the energy dispersive X-ray spectroscopy (EDS) element mapping of Co-Pi/CIS (Figure 1g), elements of Cd, In, S, Co, P, and O were simultaneously detected and evenly distributed, which demonstrates the presence of Co-Pi. 

The surface elemental composition and chemical state of the samples were measured by XPS, and the results are shown in Figure 2. The survey spectra of CIS and Co-Pi/CIS in Figure 2a reveal the elementary composition, where the signals indexed as Co and P elements in Co-Pi/CIS sample further demonstrate the successful deposition of Co-Pi on CIS surface. From the high-resolution XPS of Cd 3d in Figure 2b, there are two characteristic peaks at the binding energies of 405.4 and 412.1 eV, belonging to Cd 3d_5/2_ and Cd 3d_3/2_ of Cd^2+^, respectively [46]. The In 3d spectrum of CIS sample in Figure 2c exhibit two characteristic peaks around 445.1 and 452.7 eV, corresponding to In 3d_5/2_ and In 3d_3/2_ of In^3+^, respectively [32,47]. In S 2p spectrum (Figure 2d), signals located at 161.5 and 162.6 eV can be separately assigned to S 2p_3/2_ and S 2p_1/2_ in CIS [48]. After loading Co-Pi, in addition to the original two peaks, two more peaks appear at 162.2 and 163.3 eV. It is reasonable to infer that the Co atoms of Co-Pi easily combine with the unsaturated S atoms on CIS surface to form Co-S bonds. The Co 2p spectrum in Figure 2e shows characteristic peaks at 797.5 and 782.1 eV, which are related to Co 2p_1/2_ and Co 2p_3/2_, respectively [49]. In Figure 2f, there is a signal of P 2p at the binding energy of 133.8 eV, which is a typical characteristic peak of P in phosphate, proving that P exists in the form of PO_4_^3−^ [50]. The above results confirm the successful formation of Co-Pi/CIS heterostructure, as well as the interaction between Co-Pi and CIS through Co-S bonds.

Photocatalytic activity of CIS and Co-Pi/CIS under visible light (λ > 420 nm) was evaluated by H_2_ evolved from deionized water containing 20 vol% methanol as a hole sacrifice. Under irradiation for 2.5 h, 6.55 mmol g^−1^ H_2_ is produced over CIS (Figure 3a). Loading 3 wt% Co-Pi on CIS surface increases the photocatalytic H_2_ production amount to 18.20 mmol g^−1^ in the same period of time, which is 2.78 times that of the pristine CIS. For comparison, Pt nanoparticles with the same mass fraction were loaded on CIS surface by photochemical methods. The H_2_ production over 3% Pt/CIS sample is 9.33 mmol g^−1^ for 2.5 h. Obviously, Pt does not improve the photocatalytic performance of CIS as much as Co-Pi, probably because Pt is unable to accelerate the holes migration from the catalyst bulk to the surface. This result demonstrates that the Co-Pi/CIS system without precious metals exhibits superior catalytic activity than traditional Pt modified CIS photocatalysts. We further explored the effect of Co-Pi loading on H_2_ production, and the results are displayed in Figure 3b. With the increase in Co-Pi loading amount, the photocatalytic performance of CIS gradually improves, and the 3 wt% Co-Pi/CIS achieves the optimal H_2_ generation rate of 7.28 mmol g^−1^ h^−1^. However, when it comes to 4 wt%, the H_2_ production rate over Co-Pi/CIS reduces in turn. On the one hand, the excessive Co-Pi inevitably shields the incident light, resulting in a decrease in photoexcitation of CIS. On the other hand, the excessive Co-Pi may cover the reduction reaction sites on CIS surface, reducing the HER ability [51]. Moreover, as an important indicator for sulfide photocatalysts, the durability of Co-Pi/CIS is investigated (Figure 3c). After consecutive 10 h photocatalytic HER cycle tests, the H_2_ production performance of Co-Pi/CIS is basically maintained, indicating its excellent photostability. Figure 3d is the AQE of Co-Pi/CIS at different wavelength, showing a similar trend with light absorption. Under 405 nm monochromatic light illumination, the AQE reaches up to 14.08%. This means that the visible light excitation of CIS component plays a dominant role in the photocatalytic activity. By comparing with other metal sulfide-based photocatalysts used for H_2_ evolution, it is found that the Co-Pi/CIS in the present work exhibits superior photocatalytic performance (Table 1).

The absorption spectra of CIS and Co-Pi/CIS were derived from UV-vis DRS, which characterizes the optical properties of the samples. As shown in Figure 4a, the examined samples exhibit similar absorption band edges at about 530 nm, indicating that the modification of Co-Pi does not change the band structure of CIS. Compared to the pristine CIS, Co-Pi/CIS possesses enhanced spectra absorption in the range of 500–800 nm, which is consistent with the color of the samples. Based on the Tauc curve, the band gap of the CIS is estimated to be 2.33 eV (Figure 4b). In Figure 4c, CIS exhibits typical characteristics of n-type semiconductors due to the positive slope of the M-S plot. It is well known that the flat band potential (V_fb_) of n-type semiconductors is approximately 0.1 V lower than the conduction band potential (E_CB_) [60]. Therefore, the conduction band minimum (CBM) potential of CIS is estimated to be −0.87 V vs. reversible hydrogen electrode (RHE). Combined with the bandgap width of the CIS, the valence band maximum (VBM) position was determined to be 1.46 V vs. RHE. After coupling with Co-Pi, the flat band potential of Co-Pi/CIS drops to −0.62 V, indicating that the Fermi level of Co-Pi is lower than that of CIS. Based on the above results, the band structure diagram of the Co-Pi/CIS heterojunction can be depicted (Figure 4d). When Co-Pi is in situ deposited on CIS surface, Co-S bonds are constructed as bridges for their tight connection, and the Fermi level is aligned to the same level (−0.62 V vs. RHE). Subsequently, a built-in electric field pointing from CIS to Co-Pi is formed to promote the migration of photogenerated holes from CIS to Co-Pi.

The production of superoxide radicals ($•O_{2}^{-}$) and hydroxyl radicals (•OH) during the photocatalytic reaction was monitored by EPR, and the reasons for the enhanced photocatalytic performance of Co-Pi/CIS were discussed. 5,5-Dimethyl-1-pyrroline-N-oxide (DMPO) was used to capture $•O_{2}^{-}$ and •OH to produce complexes of DMPO-$•O_{2}^{-}$ and DMPO-•OH in methanol/aqueous solutions. The EPR signals of DMPO-•OH from photogenerated holes and DMPO-$•O_{2}^{-}$ from electrons can be used to characterize the number and state of holes and electrons on the sample surface, respectively. Compared to the pristine CIS, Co-Pi/CIS system produces much more holes under light excitation, resulting in intensified DMPO-•OH signal in EPR spectra (Figure 5a). On the other hand, the DMPO-$•O_{2}^{-}$ signal detected in Co-Pi/CIS system is not significantly improved compared to CIS (Figure 5b). These results suggest that Co-Pi promotes the photocatalytic H_2_ production performance of CIS by providing oxidation rather than reductive reaction sites and that it has a fairly strong hole extraction ability as a hole acceptor.

The effect of Co-Pi co-catalysts on the photogenerated charge transfer behavior of CIS was further verified by PL, TRPL, and I-t tests. As shown in Figure 6a, the PL emission intensity of CIS is higher than that of Co-Pi/CIS. This PL fluorescence quenching that occurs in Co-Pi/CIS systems means efficient charge injection from CIS to Co-Pi. TRPL is a powerful tool for exploring the dynamics of light-induced charge carrier transfer (Figure 6b), where τ_i_ and A_i_ represent fluorescence lifetime and amplitude, respectively. The decay kinetics of Co-Pi/CIS shows a longer average emission lifetime (68.9 ns) than CIS (63.9 ns) due to the presence of Co-Pi co-catalyst as a hole acceptor. A photoelectrode is prepared by coating the samples on FTO substrate for photocurrent–time response curve measurement. The significantly enhanced photocurrent density in the Co-Pi/CIS electrode strongly illustrates the accelerated electron–hole pair transfer kinetics (Figure 6c).

Figure 7a, b depicts the photogenerated charge transfer process in CIS and Co-Pi/CIS, respectively. Under visible-light irradiation, electrons in valence band (VB) of CIS are excited to the conduction band (CB), leaving the same number of holes in VB. The photogenerated electrons then migrate to the unsaturated S atoms on CIS surface, reacting with the adsorbed H^+^ at the active site for H_2_ production [43]. Meanwhile, due to the slow hole migration rate in CIS, it is easy for holes to recombine with electrons during their migration, ultimately leading to low charge separation efficiency and poor HER activity. Therefore, the introduction of Pt co-catalyst has a very limited effect on the photocatalytic performance improvement of CIS. For Co-Pi/CIS photocatalyst, the Co atoms of Co-Pi combine with unsaturated S atoms of CIS to form Co-S bonds. The holes generated in CIS are rapidly extracted by Co-Pi through Co-S bonds, followed by rapid consuming by methanol. The recombination of electrons and holes during migration process decreases. As a result, more electrons are able to migrate to the surface of the catalyst to participate in HER process, thereby greatly improving the photocatalytic H_2_ evolution activity of CIS.

## 4. Conclusions

In this work, cobalt phosphate (CoH_x_PO_y_, Co-Pi), a commonly used OER co-catalyst, was employed to replace Pt for photocatalytic HER activity enhancement. Using a simple photochemical method, amorphous Co-Pi nanoparticles were uniformly deposited on the surface of CdIn_2_S_4_ (CIS) nanosheets, where the Co atoms of Co-Pi preferentially bond with the unsaturated S atoms of CIS to form Co-S bonds. The optimized 3% Co-Pi/CIS sample showed a satisfactory visible-light-driven photocatalytic H_2_ production rate of 7.28 mmol g^−1^ h^−1^, which was 2.78 times that of the pristine CIS. Moreover, as a cheap and abundant non-noble metal co-catalyst, Co-Pi exhibited a better HER improvement effect on CIS photocatalyst than Pt nanoparticles. The photocatalysis mechanism of Co-Pi/CIS was systematically studied through a series of spectroscopic and photoelectrochemical characterization. As an OER promotor, Co-Pi extracted the photogenerated holes to reduce the recombination of electrons and holes during their migration from the bulk phase to the surface, which enables more electrons to participate in the HER on CIS surface. These results may pave a reasonable pathway for excitation of the photocatalytic activity of ternary metal sulfides.

## Figures and Tables

**Figure 1 nanomaterials-13-00420-f001:**
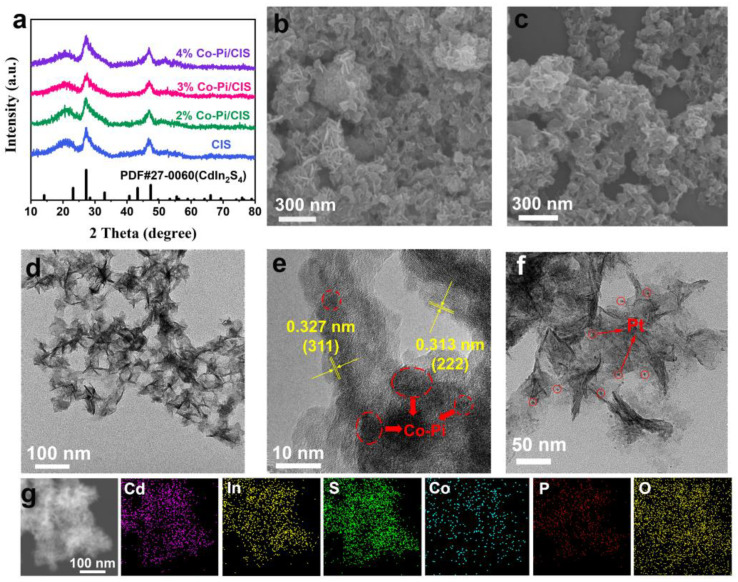
(**a**) X-ray diffraction (XRD) patterns of CdIn_2_S_4_ (CIS) and x% CoH_x_PO_y_/CdIn_2_S_4_ (Co-Pi/CIS). scanning electron microscopy (SEM) images of (**b**) CIS and (**c**) Co-Pi/CIS. (**d**) transmission electron microscopy (TEM) and (**e**) high-resolution TEM (HRTEM) images of Co-Pi/CIS. (**f**) TEM image of Pt/CIS. (**g**) TEM image and corresponding energy dispersive X-ray spectroscopy (EDS) elemental mapping of Co-Pi/CIS.

**Figure 2 nanomaterials-13-00420-f002:**
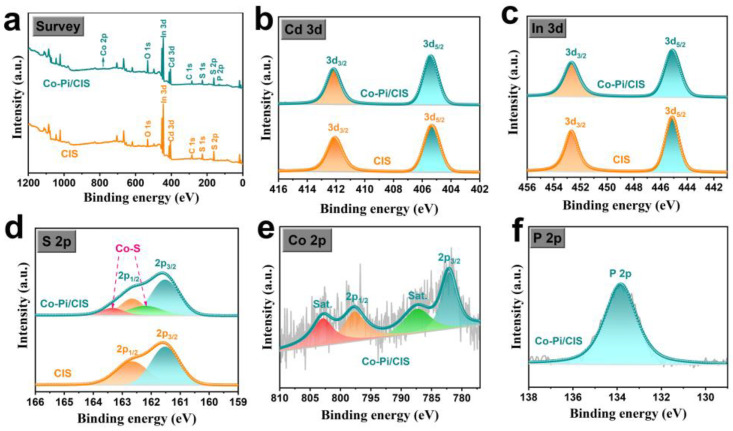
X-ray photoelectron spectroscopy (XPS) spectra of CIS and Co-Pi/CIS: (**a**) survey, (**b**) Cd 3d, (**c**) In 3d, (**d**) S 2p, (**e**) Co 2p, and (**f**) P 2p.

**Figure 3 nanomaterials-13-00420-f003:**
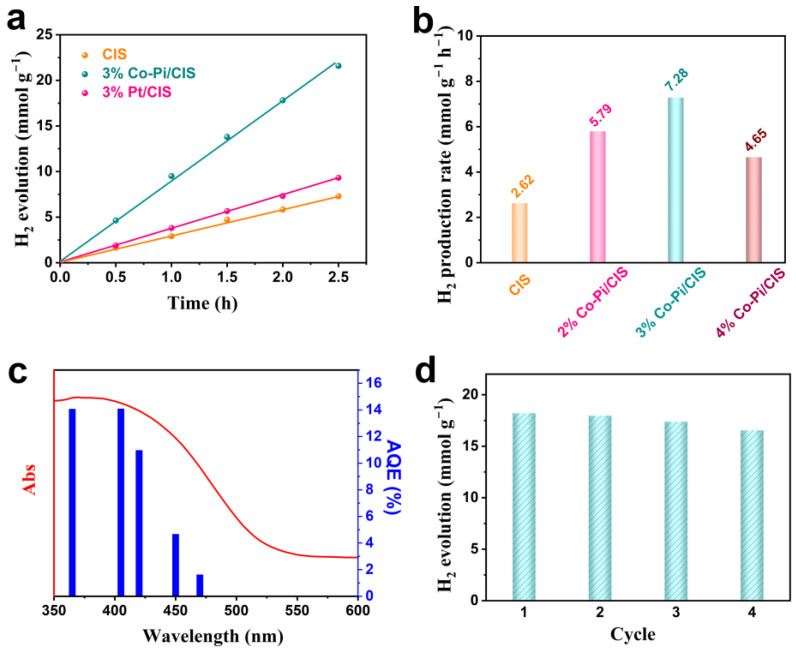
(**a**) Photocatalytic H_2_ evolution over CIS, Co-Pi/CIS and Pt/CIS under visible light irradiation, (**b**) photocatalytic H_2_ evolution rate of CIS and x% Co-Pi/CIS, (**c**) wavelength dependence of apparent quantum efficiency (AQE) and the absorption spectra of Co-Pi/CIS, and (**d**) cyclic photocatalytic H_2_ production property of Co-Pi/CIS.

**Figure 4 nanomaterials-13-00420-f004:**
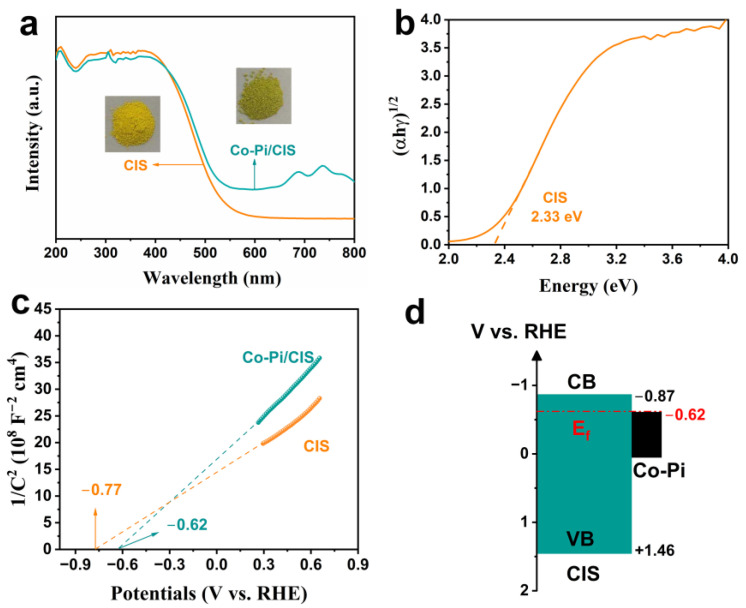
(**a**) Ultraviolet-visible diffuse reflection spectra (UV-Vis DRS) and photograph, (**b**) Tauc plot, and (**c**) Mott-Schottky (M-S) plot of the samples. (**d**) Band alignment of Co-Pi/CIS heterostructure.

**Figure 5 nanomaterials-13-00420-f005:**
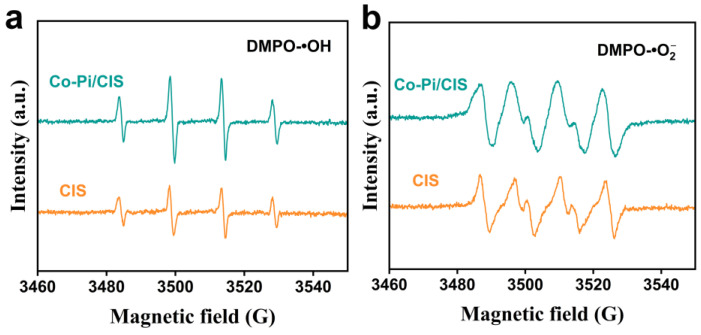
Electron paramagnetic resonance (EPR) spectra of (**a**) hydroxyl radicals (•OH) and (**b**) superoxide radicals $(•O_{2}^{-}$) trapped by 5,5-Dimethyl-1-pyrroline-N-oxide (DMPO) in the CIS and Co-Pi/CIS dispersion solution.

**Figure 6 nanomaterials-13-00420-f006:**
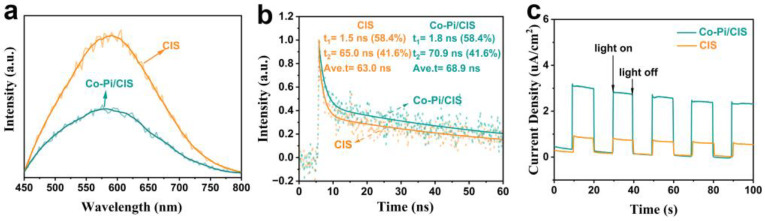
(**a**) Steady-state photoluminescence (PL) spectra, (**b**) time-resolved PL (TRPL) decay, (**c**) transient photocurrent-time (I-t) plots of CIS and Co-Pi/CIS.

**Figure 7 nanomaterials-13-00420-f007:**
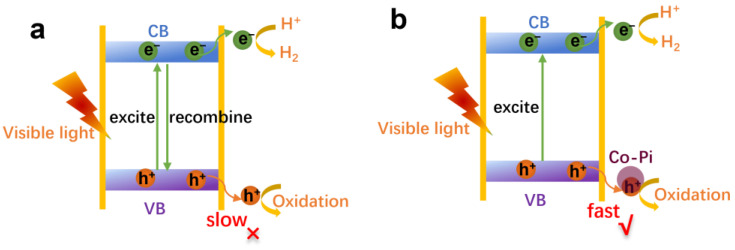
Photogenerated charge transfer process of (**a**) CIS, and (**b**) Co-Pi/CIS.

**Table 1 nanomaterials-13-00420-t001:** Comparison of representative sulfide photocatalysts and their H_2_ evolution behaviors.

Photocatalyst	Condition	Rate (mmol g^−1^ h^−1^)	AQE (%)	Ref.
Co-Pi/CIS	Methanol (λ > 420 nm)	7.28	14.08	this work
SnS/g-C_3_N_4_	lactic acid (Solar-simulator)	0.82	0.55	[52]
MoS_2_/ZnIn_2_S_4_	TEOA (λ > 420 nm)	0.22	11.8	[53]
Co_9_S_8_@CdIn_2_S_4_	Na_2_S and Na_2_SO_3_ (Solar-simulator)	4.64	13.59	[54]
CdS/CdIn_2_S_4_	Na_2_S and Na_2_SO_3_ (λ > 420 nm)	0.82	/	[55]
Au@CaIn_2_S_4_	Na_2_S and Na_2_SO_3_ (λ > 420 nm)	4.54	/	[56]
MoP/CdIn_2_S_4_	lactic acid (λ > 420 nm)	0.29	0.32	[36]
WO_3−x_/PbS	Na_2_S and Na_2_SO_3_ (λ > 420 nm)	0.64	1.5	[57]
NiS/ZnIn_2_S_4_	Na_2_S and Na_2_SO_3_ (λ > 420 nm)	3.65	0.65	[58]
CdS-SV@CuS	Lactic acid (λ > 420 nm)	1.65	6.51	[59]

## Data Availability

Data supporting this study are available within the article.

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
