# Peer review of "In Situ Photodeposition of Cobalt Phosphate (CoHxPOy) on CdIn2S4 Photocatalyst for Accelerated Hole Extraction and Improved Hydrogen Evolution"

_nanomaterials, 2023, doi:10.3390/nano13030420_

Round 1

Reviewer 1 Report

This paper deals with the preparation, structural and electronic characterization and performance of a novel photocatalyst. The system is of high interest for the readership, the research is conducted correctly and extensively and the conclusions are fully supported by the presented data. I have some suggestions for minor revisions, then in my opinion this paper will be suitable for publication.

1) line 119: "The photocatalytic performance was tested *by* visible light..."

2) line 146: "micromorphology" is not the correct term, in my opinion, since TEM images reveal structures of 30-50 nm size. "morphology" or "nanostructure".

3) Figure 2. There is a mistake in spin-orbit components labels, it is "Cd3d5/2" (instead of 1/2) and "In3d5/2" (instead of 1/2).

4) lines 240, 243, 245: the superoxide radical is O2-, not O2-. Please check and correct the text (Figure 5 labels are OK).

Author Response

1) line 119: "The photocatalytic performance was tested *by* visible light..."

Response: Thank the reviewer for the careful reading and helpful comments. We have modified the sentence as follows:

The photocatalytic performance was tested under visible light by using a 500 W Xenon lamp as a light source filtered with a nominal 420 nm cutoff filter. (line 119)

2) line 146: "micromorphology" is not the correct term, in my opinion, since TEM images reveal structures of 30-50 nm size. "morphology" or "nanostructure".

Response: Thank the reviewer for beneficial suggestions. The corresponding description in the context has been modified as follows:

The nanostructure of the samples was characterized by SEM and TEM. (line 146)

3) Figure 2. There is a mistake in spin-orbit components labels, it is "Cd3d5/2" (instead of 1/2) and "In3d5/2" (instead of 1/2).

Response: Thank you for your reminding. The mistakes in Figure 2b and c has been amended.

4) lines 240, 243, 245: the superoxide radical is O2-, not O2-. Please check and correct the text (Figure 5 labels are OK).

Response: Thank the reviewer for the careful reading and helpful reminding. We have made the following modifications for these errors:

The production of superoxide radicals (·O2-) and hydroxyl radicals (•OH) during the photocatalytic reaction was monitored by EPR, and the reasons for the enhanced photocatalytic performance of Co-Pi/CIS were discussed. 5, 5-Dimethyl-1-pyrroline-N-oxide (DMPO) was used to capture ·O2- and •OH to produce complexes of DMPO-·O2- and DMPO-•OH in methanol/aqueous solutions. The EPR signals of DMPO-•OH from photogenerated holes and DMPO-·O2- from electrons can be used to characterize the number and state of holes and electrons on the sample surface, respectively. Compared to the pristine CIS, Co-Pi/CIS system produces much more holes under light excitation, resulting in intensified DMPO-•OH signal in EPR spectra (Figure 5a). On the other hand, the DMPO-·O2- signal detected in Co-Pi/CIS system is not significantly improved compared to CIS (Figure 5b). (line 245-255)

Reviewer 2 Report

The manuscript by Jiachen Xu, et al. reports on CdIn2S4 photocatalysts towards photocatalytic hydrogen generation. The authors employ cobalt phosphate to address the intrinsic drawback of CIS, which suffers from a high recombination rate. The results are interesting. However, the manuscript contains several typos and errors that require corrections. Beyond this, I have a few comments, I ask the authors to address these issues.

The label of the x-axis in Fig d, is missing. I guess it should read "cycle", and should be corrected. Also, the indexes in Fig 1 (a) in the XRD curves should be added for better reading. I suggest the authors compare their results with other semiconducting metal sulfide-based photocatalysts used for hydrogen evolution. In this regard, the following papers can be useful. 

https://doi.org/10.3390/catal12111316

https://doi.org/10.1016/j.mtsust.2022.100305

Author Response

1) The label of the x-axis in Fig d, is missing. I guess it should read "cycle", and should be corrected. Also, the indexes in Fig 1 (a) in the XRD curves should be added for better reading.

Response: Thank the reviewer for your suggestion. The label of the x-axis in Figure 3d has been corrected. And the indexes in the XRD curves (JCPDS No. 27-0060 CdIn2S4) has been marked in Figure 1a.

2) I suggest the authors compare their results with other semiconducting metal sulfide-based photocatalysts used for hydrogen evolution. In this regard, the following papers can be useful.  

https://doi.org/10.3390/catal12111316

https://doi.org/10.1016/j.mtsust.2022.100305

Response: Thank you for your beneficial suggestions. The above two papers you provided are referenced. We have compared similar photocatalysts reported in recent years in Table 1, and the supplementary description in the text is as follows:

By comparing with other metal sulfide-based photocatalysts used for H2 evolution, it is found that the Co-Pi/CIS in the present work exhibits superior photocatalytic performance (Table 1). (line 212-214)
